# Curcumin Acetylsalicylate Extends the Lifespan of *Caenorhabditis elegans*

**DOI:** 10.3390/molecules26216609

**Published:** 2021-10-31

**Authors:** Lei Zhou, Jin Liu, Lan-Lan Bu, Duan-Fang Liao, Shao-Wu Cheng, Xi-Long Zheng

**Affiliations:** 1Key Laboratory of Hunan Province for Integrated Traditional Chinese and Western Medicine on Prevention and Treatment of Cardio-Cerebral Diseases, College of Integrated Chinese and Western Medicine, Hunan University of Chinese Medicine, Changsha 410208, China; ilsummer@aliyun.com (L.Z.); lj_lyy@163.com (J.L.); 2Division of Stem Cell Regulation and Application, Key Laboratory for Quality Evaluation of Bulk Herbs of Hunan Province, Hunan University of Chinese Medicine, Changsha 410208, China; bulanlan216@163.com (L.-L.B.); dfliao@hnucm.edu.cn (D.-F.L.); 3Departments of Biochemistry & Molecular Biology and Physiology & Pharmacology, Libin Cardiovascular Institute, Cumming School of Medicine, University of Calgary, Calgary, AB T2N 4Z6, Canada

**Keywords:** curcumin acetylsalicylate, *Caenorhabditis elegans*, aging, antioxidation

## Abstract

Aspirin and curcumin have been reported to be beneficial to anti-aging in a variety of biological models. Here, we synthesized a novel compound, curcumin acetylsalicylate (CA), by combining aspirin and curcumin. We characterized how CA affects the lifespan of *Caenorhabditis elegans* (*C. elegans*) worms. Our results demonstrated that CA extended the lifespan of worms in a dose-dependent manner and reached its highest anti-aging effect at the concentration of 20 μM. In addition, CA reduced the deposition of lipofuscin or “age pigment” without affecting the reproductivity of worms. CA also caused a rightward shift of *C. elegans* lifespan curves in the presence of paraquat-induced (5 mM) oxidative stress or 37 °C acute heat shock. Additionally, CA treatment decreased the reactive oxygen species (ROS) level in *C. elegans* and increased the expression of downstream genes superoxide dismutase *(sod)-3*, glutathione S-transferase *(gst)-4*, heat shock protein *(hsp)-16.2,* and catalase-1 *(ctl-1)*. Notably, CA treatment resulted in nuclear translocation of the DAF-16 transcription factor, which is known to stimulate the expression of SOD-3, GST-4, HSP-16, and CTL-1. CA did not produce a longevity effect in *daf-16* mutants. In sum, our data indicate that CA delayed the aging of *C. elegans* without affecting reproductivity, and this effect may be mediated by its activation of DAF-16 and subsequent expression of antioxidative genes, such as *sod-3* and *gst-4*. Our study suggests that novel anti-aging drugs may be developed by combining two individual drugs.

## 1. Introduction

Aging is a natural process featuring a gradual degeneration in physiological integrity [1] and is connected with many chronic diseases, such as neurodegenerative disease, cardiovascular diseases, diabetes, etc. [1,2]. Aspirin (acetylsalicylic acid HC_9_H_7_O_4_), a cyclooxygenase inhibitor, is a nonsteroidal anti-inflammatory drug (NSAID) discovered over a century ago and has been widely used in the treatment of conditions such as aches, fever, and inflammation. Aspirin inhibits oxidant generation and glycoxidation reactions, the main risk factors for aging. Low-dosage administration of aspirin has an anti-aging effect likely due to its protection of cellular and functional declines, particularly from inflammatory and oxidative sources [3], which are known to contribute to aging. Treatment with aspirin has been reported to positively affect the anti-aging of male mice [4,5]. Aspirin also extends the lifespan of *Caenorhabditis*
*elegans* (*C. elegans*) and strengthens their stress resistance through a mechanism involving DAF-16/FOXO, AMPK, and LKB1 [5,6]. Aspirin induces autophagy through its competitive inhibition of the acetyltransferase activity of EP300 or the EP300 ortholog *cpb-1* in *C. elegans* and increases lifespan by acting as a caloric-restriction mimetic in various mouse models [6,7]. Long-term use of aspirin has many side effects, including gastrointestinal bleeding or cerebral hemorrhage and kidney failure [8,9]. Thus, many aspirin derivatives have been synthesized and tested for their lifespan extension effects in *C. elegans* [10,11].

Curcumin is a polyphenolic compound and has anti-aging effects in nematodes, fruit flies, and mice [12,13,14,15], likely through its antioxidant and anti-inflammatory effects. For example, curcumin delays cell aging in *Saccharomyces cerevisiae* (*S. cerevisiae*) [16]. Curcumin extends the lifespan of worms by reducing intracellular reactive oxygen species (ROS) and lipofuscin [12] or “age pigment” deposition during aging. The mechanisms underlying the curcumin anti-aging effect in *C. elegans* involves its antioxidative properties, which affect the size and the pharyngeal pumping rate, suggesting a mechanism similar to dietary restriction. Notably, curcumin does not have a lifespan extension effect in *osr-1*, *sek-1*, *mek-1*, *skn-1*, *unc-43*, *sir-2.1*, and *age-1* mutants, suggesting the roles of these genes in the curcumin effect. Importantly, curcumin maintains a lifespan prolonging effect in *mev-1* and *daf-16* mutants [12], suggesting a differential mechanism involved. Notably, low absorption and poor bioavailability of curcumin have largely limited its clinical application. Studies have reported that curcumin is structurally unstable and may have false-positive results in vitro and in vivo [17,18], but some scholars believe that curcumin should not be completely rejected [17].

To address these issues, we introduced the aspirin group into curcumin and esterified two phenolic hydroxyl groups in its structure to obtain a new drug, curcumin acetylsalicylate (CA, patent application number: CN2012101010942A). The optimum synthesis conditions of CA compound (C_39_H_32_O_12_), in which MS(ESI, M/Z):[M-H]+ is 693, were as follows: n (curcumin):n (aspirin):n (dichlorosulfide):n (pyridine) = 1:6:5.5:6, reaction temperature −5 °C, reaction solution pH 5–6. The yield and content of the product reached 57.2% and 99.3% after the silica gel column [19]. Our preliminary studies have revealed various pharmacological effects of CA both in vivo and in vitro, including antioxidative, anti-inflammatory, and vasculoprotective effects. In the current study, we characterized CA effects on the lifespan of worms. Our results showed that CA has a potent anti-aging effect and anti-oxidation ability.

## 2. Materials and Methods

### 2.1. Chemicals and Materials

Curcumin acetylsalicylate (CA, mass fraction > 98%) was synthesized by acyl chloride reaction in the College of Pharmacy, Hunan University of Chinese Medicine, China [19]. Rapamycin from MedChemExpress LLC (Beijing, China), paraquat-dichloride (C_36_H_32_F_24_N_4_P_4_), and levamisole hydrochloride were purchased from Solarbio Life Sciences (Beijing, China). Rapamycin or CA was dissolved in dimethyl sulfoxide (DMSO) with final concentrations of less than 0.1% in all experiments, which did not have any significant effect on the lifespan of worms [20]. Paraquat and levamisole hydrochloride were dissolved in distilled water. The ROS assay kit and the dichloro-dihydro-fluorescein diacetate (DCFH-DA) were purchased from Beyotime Biotechnology (Shanghai, China).

### 2.2. C. elegans Strains and Maintenance

Experimental nematodes were kept at 20 °C, unless otherwise stated, on nematode growth media (NGM) plates seeded with Escherichia coli OP50 [21]. All *C. elegans* strains were provided by the Caenorhabditis Genetics Center (CGC, University of Minnesota, Minneapolis, MN, USA), which included wild-type N2, CF1553 (mu1s84[pAD76(*sod*-3::GFP)]), TJ356 (zIs356[*daf-16p::daf-16a/b*::GFP *+ rol-6*]), CL2166 (dvIs19[pAF15(*gst-4::*GFP::NLS)]), and CF1038 *[daf-16(mu86) I.]*.

### 2.3. Lifespan Assay

The lifespan of *C. elegans* was determined according to the experimental method of Dilberger B et al. [22]. In brief, worms were kept on NGM OP50 plates at 20 °C and synchronized to the L4 stage by timed egg-laying. 25 worms with a large number of eggs were treated with bleaching solution. When the eggs grew to the peak period of egg-laying, some worms were selected to new NGM plates to lay eggs for 1–3 h, and then the adults were picked out to leave only the eggs, achieving the effect of synchronous egg-laying. When these eggs hatched and the larvae grew to the L4 stage, they were taken out for experiments. Synchronized L4 N2 or *daf-16(mu86)* worms (≥60 worms per plate) were transferred to NGM plates with DMSO (control) or with 2 nM rapamycin (positive control) [23] or different concentrations of CA (5, 10, 20 μM). Each group contained 3 NGM plates, with the setup time as day 0. The survival status of *C. elegans* was recorded daily until they all died. Death was recorded when the worms did not react to light stroking with a picker and their pharyngeal pump stopped pumping; they were then scored and removed. Experiments were repeated 3 times with at least 165 worms in each group.

### 2.4. Stress Resistance and Heat Shock Assays

The culture and grouping of worms were the same as above. Refer to the study of Yang ZZ et al. [24], with minor modifications based on preliminary experimental results. After 5 d culture, worms were tested for their stress resistance. In the oxidative stress resistance experiment, worms treated with various compounds for 5 d were transferred to new NGM OP50 plates (including 5 mM paraquat). Paraquat, as a stressor of the mitochondrial respiratory chain, was used to induce strong oxidative stress and simulate premature senescence conditions in worms [22]. The survival status of worms was scored every 12 h. In the heat shock assay, worms after 5 d treatment at 20 °C were transferred to 37 °C [22,24], followed by scoring for death hourly until all nematodes died.

### 2.5. Lipofuscin Assay

Synchronized L4 worms were transferred to NGM OP50 plates with DMSO (control) or with rapamycin (2 nM) or CA (5, 10, 20 μM) for 7 d at 20 °C [25]. The worms were then anesthetized with 10 mM levamisole hydrochloride and fixed on 2% agarose slides [26]. The blue autofluorescence of lipofuscin was observed using an inverted fluorescence microscope with a 10× objective lens (*λ*ex470/40 nm, *λ*em525/50 nm, Carl Zeiss, Axio Vert.A1, Gottingen, Germany). ImageJ software was used to measure the relative fluorescence intensity of each nematode after removing background fluorescence, and GraphPad software was used for semi-quantitative analysis [27,28].

### 2.6. Reproduction Assay

Synchronized wild-type L4 worms were transferred to NGM OP50 plates with DMSO (control) or with rapamycin (2 nM) or CA (5, 10, 20 μM) with 10 plates for each condition (1 worm/plate). Worms were transferred to new plates daily to distinguish between parents and progeny until the end of oviposition [29]. Eggs in the NGM OP50 plates were kept in the incubator at 20 °C and the number of progeny was recorded when they reached the L3 stage. All experiments were repeated at least twice.

### 2.7. ROS Measurement

Synchronized L4 worms were transferred to NGM OP50 plates with DMSO (control) or with rapamycin (2 nM) or CA (5, 10, 20 μM) for 5 d with 65 worms in each condition (triplicate). Nematodes were then transferred to new plates, with or without (blank control) 5 mM paraquat for 24 h [30], followed by washing with M9 buffer. The ROS content was then measured according to the ROS assay kit instructions. The worms were transferred to a 96-well plate (50 worms per well) containing 50 μM DCFH-DA and incubated at 20 °C for 30 min. Excessive DCFH-DA was removed using M9 buffer, followed by detection of the total fluorescence in each well on a Biotek microplate reader or observation with a fluorescent microscope (*λ*ex488 nm, *λ*em525 nm). Fluorescence was quantified using ImageJ software.

### 2.8. Nuclear Translocation of DAF-16

The transgenic strain TJ356 expressing DAF-16::GFP was used for the detection of DAF-16 transcription factor nuclear localization, as previously described by Yuan Y et al. [28]. Minor modifications were made based on our preliminary experimental results. In brief, worms synchronized to the L4 stage were transferred to NGM OP50 plates with DMSO (control) or with CA (5, 10, 20 μM) or rapamycin (2 nM) for 5 d. Worms were then transferred to NGM OP50 plates containing 5 mM paraquat for 24 h, followed by an M9 buffer solution. Worms were then anesthetized with 10 mM levamisole hydrochloride and fixed on 2% agarose slides (25 individuals per group). The entry of DAF-16 into the nuclei was observed and imaged by fluorescence microscopy with a 10x objective lens (*λ*ex530/40 nm, *λ*em575–640 nm) and the distribution of cytoplasm, intermediate, and nucleus was statistically analyzed.

### 2.9. Visualization of SOD-3::GFP and GST-4::GFP

Synchronized transgenic worms (L4 larva), including the CF1553 strain containing SOD-3::GFP reporter and the CL2166 strain containing GST-4:GFP reporter, were treated with DMSO (control) or with rapamycin (2 nM) or different concentrations of CA for 3 d with 60 worms in each group, followed by measuring total fluorescence as previously described [24,28,31]. In brief, worms were then transferred to NGM OP50 plates containing 5 mM paraquat for 24 h, followed by the M9 buffer. The worms were fixed on 2% agarose slide (25 individuals per group) with 5 mM levamisole hydrochloride [32]. The expression of GFP was imaged by fluorescence microscopy with a 10× objective lens at a constant exposure time.

### 2.10. Quantitative Real-Time PCR

Synchronized wild-type worms were placed into culture without (control) and with rapamycin (2 nM) or CA (5, 10, 20 μM) for 5 d and then placed into NGM OP50 plates containing 5 mM paraquat for 24 h [24]. Each group contained three parallel plates with 100 worms in each. The worms were flushed with M9 buffer, and 250–300 worms were picked into Eppendorf tubes containing 500 μL Trizol and flash-frozen in liquid nitrogen. Total RNA was extracted using Transzol Up Plus RNA Kit (TransGen Biotech, Beijing, China). RNA concentration was measured using a Cytation 5 imaging reader (BioTek instruments, Winooski, VT, USA) at an absorbance of 260 and 280 nm. The cDNA reverse transcription of 1 μg RNA was performed using a TransScript^®^ Reagent Kit with a gDNA eraser (AT344, Transgen Biotech). The quantitative real-time PCR was performed with SYBR Green Super-mix (TransGen Biotech, Beijing, China) in a MiniOption™ Real-Time PCR Detection System (Bio-Rad Laboratories, Inc., Hercules, CA, USA). The primer pairs used for qPCR were as follows: *daf-16*, forward 5′-CCAGACGGAAGGCTTAAACT-3′, reverse 5′-ATTCGCATGAAACGAGAATG-3′; *skn-1*, forward 5′-AGTGTCGGCGTTCCAGATTTC-3′, reverse 5′-GTCGACGAATTGCGAATCA-3′; *hsp-16.2*, forward 5′-CTGCAGAATCTCTCCATCTGAGTC-3′, reverse 5′-CTGCAGAATCTCTCCATCTGAGTC-3′; *ctl-1*, forward 5′-CGGATACCGTACTCGTGATGAT-3′, reverse 5′-CCAAACAGCCACCCAAATCA-3′; *act-1*, forward 5′-TGACGGACAAGTCATCACCG-3′, reverse 5′-CATGGTGGTTCCTCCGGAAA-3′. The expression levels of each mRNA relative to the *act-1* gene as a reference gene were calculated with the comparative 2^−ΔΔCT^ method.

### 2.11. Statistical Analysis

Statistical analyses were performed using GraphPad Prism 8.0. Data were expressed as the mean ± standard error of the mean (SEM). Survival analysis was conducted using the Kaplan-Meier method, and statistical significance was analyzed by a log-rank (Mantel-Cox) test. The fluorescence intensity of GFP was quantified as mean pixel density by ImageJ software. Differences between the control and treatment groups were determined by either the student’s *t*-test or one-way ANOVA. Values of *p* < 0.05 were considered significant.

## 3. Results

### 3.1. CA Promotes the Lifespan of C. elegans

We first determined the lifespan of *C. elegans*, the most direct and convincing indicator of longevity, without (control with the same volume of DMSO) and with CA treatment at different dosages (10, 20, or 40 μM). As shown in Figure 1B and Table 1, CA at 20 μM had the best effect, followed by 10 μM, but it no longer had any anti-aging effect when CA concentration reached 40 μM. We then adjusted the concentrations of CA to 5, 10, and 20 μM for further experiments. We also used rapamycin (2 nM) as a positive control, which is known to significantly increase lifespan [33], as previously described by Zhihui Bai et al. [23]. As expected, treatment with rapamycin increased lifespan. Importantly, CA at all three concentrations tested promoted the longevity of *C. elegans* in a concentration-dependent manner (Figure 1C, Table 2).

### 3.2. CA Treatment Reduces Lipofuscin but Not Ovulation of C. elegans

Several physiological changes, including lipofuscin age pigment aggregation, and reduced reproductive capacity, are associated with aging [29]. It is known that aging results in the accumulation of lipofuscin in the gut of worms, which is considered an aging-associated indicator [32]. Our results showed that CA treatment reduced lipofuscin pigment aggregation (Figure 2A,B). Egg production was also recorded for worms at 20 °C. As shown in Figure 2C, treatment with CA at various concentrations did not significantly affect the egg-laying rates of worms when compared with either rapamycin or vehicle control (*p* > 0.05). Thus, our data suggest that CA delays the lifespan of worms because CA reduces the accumulation of lipofuscin without sacrificing reproductive ability.

### 3.3. CA Extends *C. elegans* Survival under Stress

The longevity of worms is positively correlated with their survival in response to various stressors [33]. Thus, we examined whether treatment with CA increased their longevity under oxidative stress. Our results showed that pretreatment with CA at 20 μM increased lifespan by 20% compared with the control group (Figure 3A, Table 3). Under 37 °C heat shock, the survival rate of worms at 20 μM CA was increased by 16.27% (Figure 3B, Table 4), which was significantly longer than the control worms under stress (Figure 3 and Table 3 and Table 4; *p* < 0.05).

### 3.4. CA Improves the Antioxidant Capacity of C. elegans

ROS is the most important causative factor of oxidative damage and promotes the aging of organisms, including humans [34]. Paraquat has been used for the model of premature aging in *C. elegans* since it rapidly increases ROS production and induces severe oxidative damage in worms. Our results showed that CA decreased ROS accumulation in *C. elegans* in response to treatment with 5 mM paraquat (Figure 4, *p* < 0.0001). Correspondingly, our results showed that CA treatment significantly increased SOD expression (Figure 5, *p* < 0.0001).

### 3.5. CA Upregulates SOD-3 Expression in C. elegans

*Sod-3*, one of the transcriptional targets of DAF-16, encodes mitochondrial Mn-SOD [35], a well-known free radical scavenging SOD [36]. Therefore, we determined whether CA regulated the expression of *sod-3* using the transgenic strain CF1553, which contains SOD-3::GFP. Our results showed that treatment with 20 μM CA induced a 14% increase in fluorescence intensity compared with the control under oxidative stress (Figure 5, *p* < 0.0001). These results suggest that CA-induced reduction of ROS results from its upregulation of *sod-3*.

### 3.6. CA Upregulates GST-4 Expression in C. elegans

GST-4 is a glutathione S-transferase that facilitates the Phase II detoxification process [37]. Its expression is regulated by *daf-16* and *skn-1* [38]. Using the transgenic strain CL2166 containing GST-4::GFP, we found that treatment with 20 μM CA significantly increased GST-4 expression by 92% in the presence of 5 mM paraquat stress when compared with the control group (Figure 6, *p* < 0.01).

### 3.7. CA Treatment Induces Nuclear Localization of DAF-16::GFP in the Presence of Oxidative Stress in C. elegans

DAF-16 is an important transcription factor in the *C. elegans* insulin/IGF-1 signaling pathway, which plays a vital role in lifespan regulation. As expected, our results showed that DAF-16 was normally localized in the cytosol (Figure 7A), and when activated, translocated to the nuclei and aggregated in a large dot pattern (Figure 7C). After entering the nuclei, DAF-16 regulates the transcription of various downstream genes. To determine how CA regulated DAF-16, we cultured transgenic strain TJ356, which expressed DAF-16::GFP in a wild-type background [39]. Our results showed that treatment with 20 μM CA for 5 days enhanced DAF-16 nuclear localization by 28.84% compared to the control group in the presence of paraquat-induced oxidative stress (Figure 7D), suggesting a potential role for *daf-16* in CA effects on longevity.

### 3.8. Effects of CA on Aging-Related Gene Expression in C. elegans

It is well known that the insulin/IGF-1 receptor signaling pathway plays a critical role in *C. elegans* aging. To determine whether CA exerts its effects through this mechanism, we analyzed the mRNA expression of *daf-16*, *hsp-16.2*, *ctl-1*, and *skn-1* in wild-type worms. DAF-16 and SKN-1 are important transcription factors in the insulin/IGF-1 receptor signaling pathway. Small heat shock protein-16.2 (HSP-16.2) and catalase-1 (CTL-1) are both downstream targets of DAF-16, which are mainly related to antioxidant and stress resistance [40,41]. Compared with the control group, mRNA expression levels of *skn-1*, *ctl-1*, *daf-16*, and *hsp-16.2* in the 20 μM CA group increased by 18%, 42%, 34%, and 64%, respectively (Figure 8).

### 3.9. The Roles of DAF-16 in the Regulation of C. elegans Lifespan

The above experimental results suggest that *daf-16* is involved in CA-induced aging delay. To further determine the role of *daf-16*, we repeated the survival experiments using mutant strain CF1038, which has the loss of function *daf-16* allele *mu86*. The experimental results showed that, compared with the control group, treatment with 20 μM CA did not improve the lifespan of worms (Figure 9, Table 5), suggesting that the CA effect on longevity is dependent on DAF-16.

## 4. Discussion

Our current study revealed that CA extends the natural lifespan of *C. elegans* and is more potent than either aspirin or curcumin. CA significantly extended the lifespan of wild-type worms in a dose-dependent manner under either normal conditions or heat shock or oxidative stress conditions, suggesting that CA has the potential of providing anti-aging and stress resistance with improved bioavailability. Importantly, CA reduced the accumulation of lipofuscin in worms without a significant effect on their ovulation. Notably, treatment with CA decreased ROS production and increased the expression of SOD-3 and GST-4, suggesting a potential mechanism involving antioxidative effects of CA. Supportively, CA treatment increased nuclear translocation of DAF-16, and mutation of *daf-16* abolished CA effects. Thus, we conclude that CA has an anti-aging effect on *C. elegans*.

Although research on anti-aging drugs has made rapid progress, none have been approved for clinical use. Increasing evidence indicates that *C. elegans* is an important animal model for pre-clinical studies of anti-aging drugs. Thus, our data have provided the first evidence to demonstrate the anti-aging effects of CA as a novel compound derived from curcumin and aspirin, encouraging further clinical research. Curcumin, a well-known polyphenol compound, has anti-aging effects in various model systems, including *C. elegans* [32,42,43]. Also, aspirin extends the lifespan of *C. elegans* and improves its ability to resist oxidative stress [10,32,44]. Both curcumin and aspirin have the potential to be used for human longevity. However, due to the low bioavailability of curcumin, a large dosage will be required, which causes obvious side effects, including nausea and diarrhea [45]. Long-term, high-dose use of aspirin will result in many side effects. One of its most dangerous side effects is gastrointestinal hemorrhage and cerebral hemorrhage [9], which largely limits its long-term use. Here, we combined curcumin and aspirin as a single compound, CA, which may reduce unwanted actions and maximize their anti-aging effects, but much research will be required to confirm this possibility. It is critical to note that CA has potent anti-aging effects, as we predicted. Unlike other drugs that exert anti-aging effects at the expense of reproductive ability [46,47], our results showed that egg production was not affected.

Our finding that CA delays the aging of *C. elegans* and improves their stress resistance will likely open a new avenue for the development of anti-aging drugs. Note that CA pretreatment significantly extended the lifespan of wild-type worms (N2) under normal and stress conditions, and the lifespan curve of *C. elegans* has been the gold standard used to assess the anti-aging ability of drugs [48]. Our data showing that 7-day pretreatment with CA reduced the deposition of lipofuscin further supports the anti-aging effect of CA because the accumulation of lipofuscin, a biomarker of aging [49], is inversely proportional to longevity and positively correlated with the level of oxidative stress in *C. elegans* [50]. Note that in our study, lipofuscin levels were indicated by total blue fluorescence, as described by many other studies [28,51,52]. Blue autofluorescence has been accepted as an excellent measure of the proportion of near-dead animals in a population. It was previously reported that autofluorescence in red wavelengths might be more appropriate for characterizing aging in *C. elegans,* and red autofluorescence was highly correlated with the future lifespan of each individual [53], but this approach will need to be further confirmed. In addition, our data demonstrated that the stress resistance of *C. elegans* is positively correlated with longevity induced by CA, consistent with previous studies [50,54].

Our data support an antioxidative mechanism underlying the anti-aging effects of CA. First, the results shown in Figure 4 demonstrate that CA treatment reduced ROS production in response to paraquat. Treatment with 5 mM paraquat results in the production of a large amount of ROS and thus has been widely used as a stressor in the mitochondrial respiratory chain [22]. The excessive accumulation of ROS usually leads to mitochondrial dysfunction and accelerates the process of aging [49]. Supportively, our preliminary data in animal and cell culture studies revealed that CA has a variety of biological activities, including antioxidative, anti-inflammatory, and vasculoprotective effects (data not shown), which are known to contribute to anti-aging effects.

ROS is mainly cleared by antioxidant enzymes, among which the most important are SOD-3 and GST-4 [50]. Our data have shown that CA treatment upregulated the expression of both SOD-3 and GST-4 (Figure 5 and Figure 6), suggesting that the effect of CA on ROS may result from the upregulation of these enzymes. It must be noted that the expression of both SOD-3 and GST-4 can be regulated by DAF-16 [38,50]. Indeed, we observed increased nuclear translocation of DAF-16 in response to CA treatment, as shown in the transgenic strain TJ356 (Figure 7). Our real-time PCR data showed that *daf-16*, in response to CA treatment, induces the transcription of various genes, including *ctl-1* and *hsp-16.2* (Figure 8). Note that *gst-4* is also regulated by the transcription factor SKN-1. However, compared with the control group, the upregulation of *skn-1* by CA was not statistically significant. Thus, the upregulation of GST-4 may result from DAF-16. Importantly, the role of DAF-16 was further confirmed using a *daf-16* mutant, in which CA did not demonstrate any anti-aging effects (Figure 9). In our current study, however, we did not further characterize the involvement of insulin receptor signaling, which might be tested using *daf-16* and *daf-2* mutants*. daf-2* is homologous to the mammalian insulin/insulin-like growth factor-1 (IGF-1) receptor, which regulates *daf-16* transcriptional activity. It remains untested how CA interacts with DAF-2 and what mechanism is upstream of DAF-2.

Nevertheless, the specific targets or mechanisms for CA are not yet clear as the mechanisms underlying the effects of aspirin and curcumin are still under investigation. Aspirin, an inhibitor of cyclooxygenase, is often used in mild to moderate pain and has anti-inflammatory and anti-pyretic properties. Increasing evidence supports aspirin as a potential geroprotective agent [55]. Aspirin was recently reported to extend the lifespan of *C. elegans* and mice [3,44,45]. Further mechanistic studies revealed that the extension of *C. elegans* lifespan by aspirin is through activation of downstream DAF-12 and DAF-16 [45]. Aspirin reduces ROS production, likely through gene expression of antioxidant enzymes such as catalase, superoxide dismutase, and glutathione-S-transferase [44]. It was also reported that treatment with aspirin extends the *C. elegans* lifespan and improves stress resistance in a dietary restriction-like manner [5,6] with autophagy activation and the mitochondrial unfolded protein response [11].

Notably, curcumin is widely reported to have medicinal activity and gives false signals in drug screening tests [18]. Many research papers and review articles have been published to support a role for curcumin in the treatment of various diseases, such as erectile dysfunction, baldness, cancer, and Alzheimer’s disease, but there have been no specific therapeutic benefits reported yielding a proven treatment to date. Curcumin does not bind to any specific protein but may disrupt cell membranes [56]. Curcumin may be degraded into other compounds that have different properties and associated biological activities. Many researchers arguably believe that curcumin has biological activity and may interact with many different proteins to induce differential effects on a variety of pathological processes. As such, chemically modified forms of curcumin may be able to reach specific tissues and targets.

Our current study has suggested that the mechanism underlying the CA anti-aging effect on worms was similar to that of aspirin [5]. Note that aspirin has an effect on stress response and lifespan extension along with activation of autophagy. It is, thus, possible that CA also acts through autophagy, but this possibility will need further verification. Compared with curcumin, the efficacy curve of CA showed a dose-dependent relationship, which was not demonstrated in curcumin [12], suggesting an advantage. In addition, our experimental results indicate that CA was dependent on the transcription factor DAF-16 to delay the aging of worms, which was different from the previously reported mechanism for curcumin. Liao et al. [12] reported that curcumin extended the lifespan of *daf-16* mutants, suggesting that curcumin delayed senescence independent of *daf-16* transcriptional activity.

There have been several notable limitations in our studies. First, we have not determined the underlying mechanisms in comparison with aspirin and curcumin in terms of their effects on aging. Although we demonstrated that CA reduced ROS production, we have not shown whether CA targets cytoplasmic or mitochondrial ROS. In addition, we did not measure the bioavailability of either curcumin or CA. Thus, we do not have any data to indicate whether CA has any improved bioavailability or whether it can be metabolized by the worms. Nevertheless, our results have shown that CA increases the lifespan of *C. elegans* without or with oxidative or heat shock stress, likely through the upregulation of ROS scavenger enzymes, including SOD-3 and GST-4. DAF-16 transcriptional activity may mediate CA effects. Much more investigation will also be needed about how the mechanisms underlying CA effects differ from those of curcumin and aspirin. Nevertheless, our study has opened a new avenue to develop novel anti-aging drugs.

## Figures and Tables

**Figure 1 molecules-26-06609-f001:**
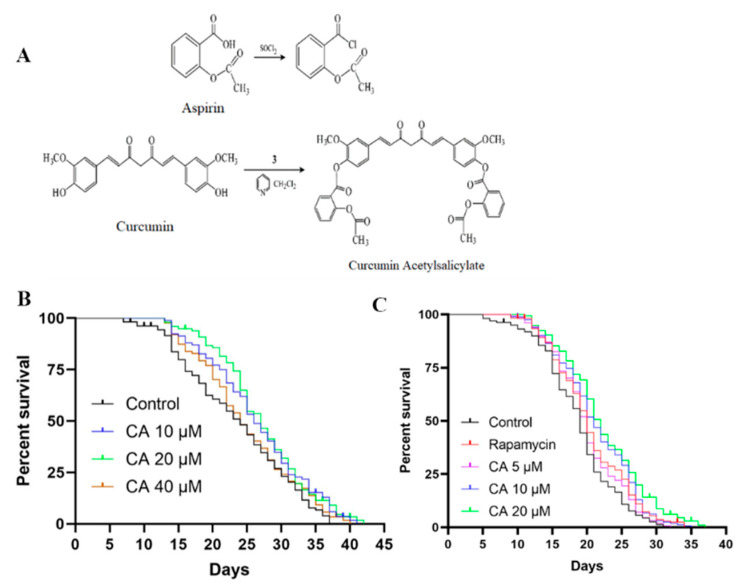
CA increases lifespan in *C. elegans*. (**A**) Structural diagram and synthesis steps of aspirin curcumin ester to form CA. Wild-type L4 stage worms were treated with 0.1% DMSO (vehicle control) or (**B**) with CA (10, 20, 40 μM, *n* > 151 in each group, *p* = 0.03, 0.02, 0.38) or (**C**) with rapamycin (2 nM) or CA (5, 10, 20 μM, *n* > 165 in each group, *p* < 0.05). Their survival was monitored from the first day of the L4 stage until death, as described in the Methods. Assays were performed for each condition at least three times. A representative trial is shown. The difference was examined with the log-rank test.

**Figure 2 molecules-26-06609-f002:**
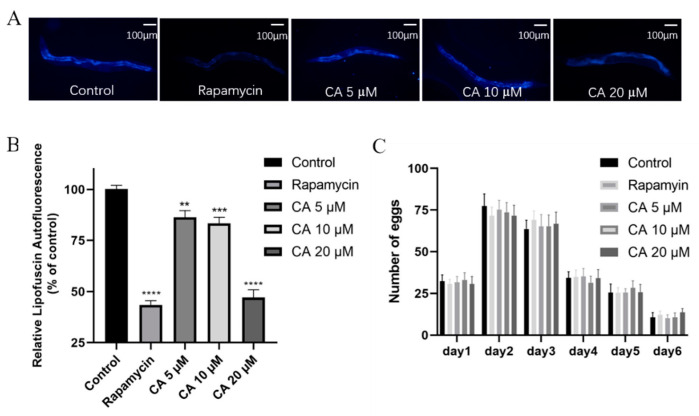
Effects of CA on lipofuscin and reproduction of *C. elegans*. Wild-type worms were treated with DMSO (control) or with rapamycin (2 nM) or CA (5, 10, 20 μM) for 5 days, followed by detection of lipofuscin autofluorescence as shown by the representative images of worms, scale bar = 100 μm. (**A**) Cumulative data are shown in the bar figure (**B**) *n* > 19 worms per group, ** *p* = 0.0016, *** *p* = 0.0001, **** *p* < 0.0001). Egg production rates were recorded (**C**) *n* = 10 worms per group, *p* > 0.05).

**Figure 3 molecules-26-06609-f003:**
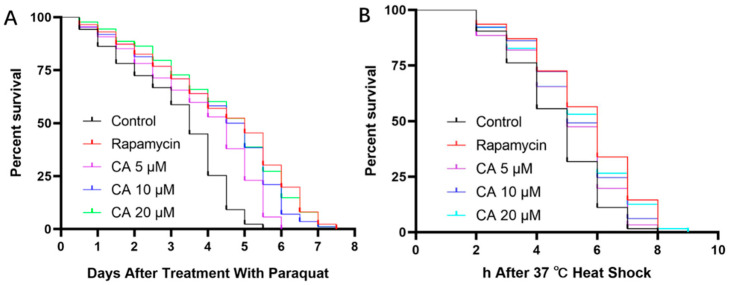
Effect of CA on *C. elegans* lifespan under stress conditions. Wild-type worms, after being synchronized to the L4 stage, were pretreated with DMSO (control) or with rapamycin (2 nM) or CA (5, 10, 20 μM) for 5 days, followed by exposure to 5 mM paraquat (**A**) *n* = 86–90, *p* < 0.05) or 37 °C heat shock (**B**) *n* = 62–66, *p* < 0.05).

**Figure 4 molecules-26-06609-f004:**
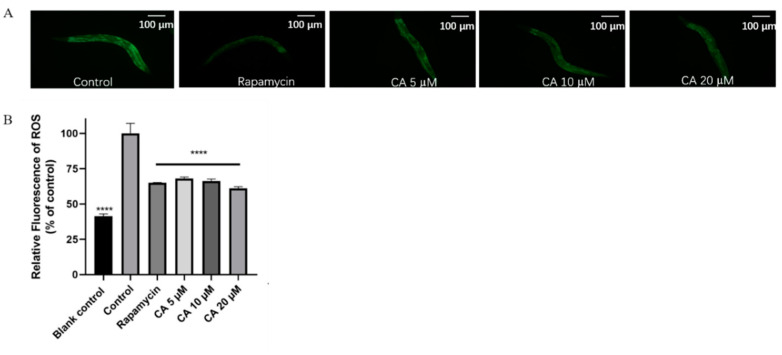
Effect of CA on intracellular ROS accumulation in *C. elegans*. Representative fluorescence images of worms, scale bar = 100 μm. (**A**) Relative fluorescence levels of ROS under oxidative stress compared with that in control (**B**) *n* > 150, **** *p* < 0.0001). Data were obtained from the Biotek microplate reader.

**Figure 5 molecules-26-06609-f005:**
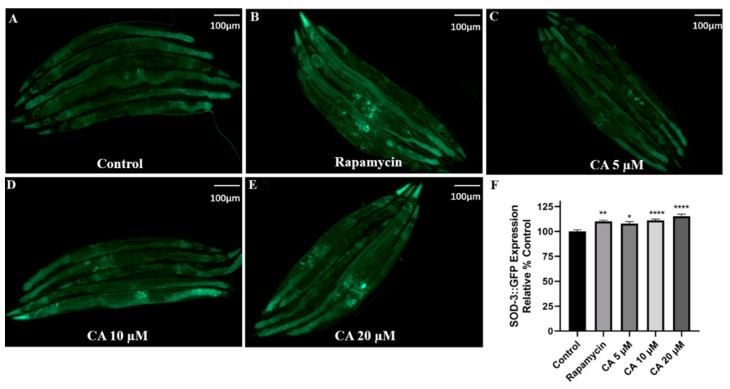
CA upregulates SOD-3 expression in *C. elegans*. Transgenic worms harboring SOD-3::GFP were treated without (control) or with rapamycin (2 nM) or CA (5, 10, 20 μM) for 3 days. Representative images (**A**–**E**) of SOD-3::GFP expression in worms with different treatments, scale bar = 100 μm. Quantification of SOD-3::GFP fluorescence intensity (**F**, *n* = 25, **** *p* ≤ 0.0001, ** *p*≤ 0.01, * *p* ≤ 0.05 vs. control).

**Figure 6 molecules-26-06609-f006:**
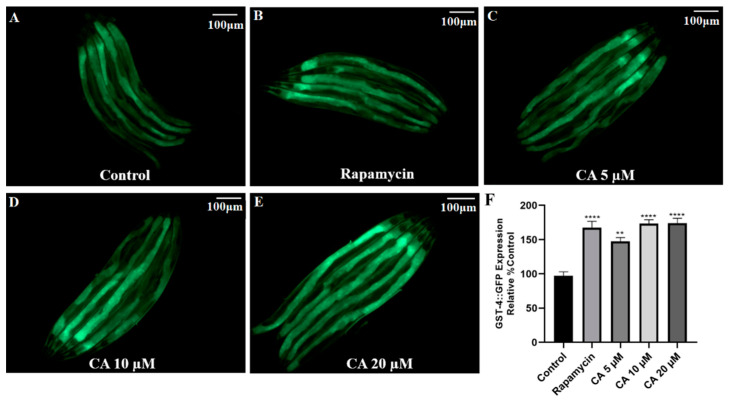
Effect of CA on GST-4 expression in *C. elegans*. Transgenic worms harboring GST*-*4::GFP were treated with DMSO (control) or with rapamycin (1 nM) or CA (5, 10, 20 μM) for 3 days, followed by imaging with fluorescence microscopy. (**A**–**E**) Representative images of GST*-*4::GFP expression. (**F**) Quantification of GST-4::GFP fluorescence intensity in worms with different treatments as indicated (*n* = 25 worms per group). The difference was examined with one-way ANOVA with Dunnett’s multiple comparisons test, **** *p* ≤ 0.0001, ** *p* ≤ 0.01 vs. control, scale bar = 100 μm.

**Figure 7 molecules-26-06609-f007:**
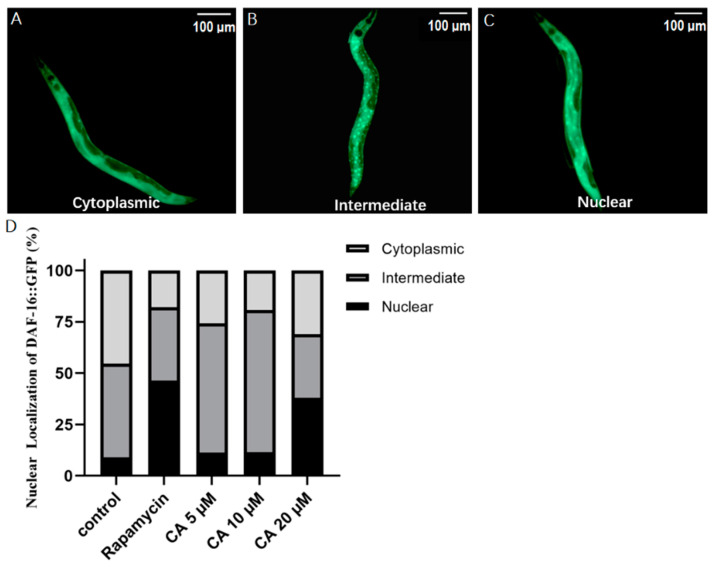
Effect of CA on DAF-16 localization in *C. elegans*. Transgenic worms harboring DAF-16::GFP were treated with DMSO (control) or with rapamycin (1 nM) or CA (5, 10, 20 μM) for 5 days. Representative images of DAF-16::GFP expression (**A**–**C**), scale bar = 100 μm. (**D**) Quantitative data of DAF-16::GFP fluorescence in different regions. Statistics of DAF-16 nucleation translocation (*n* = 25 worms per group, *p* < 0.05).

**Figure 8 molecules-26-06609-f008:**
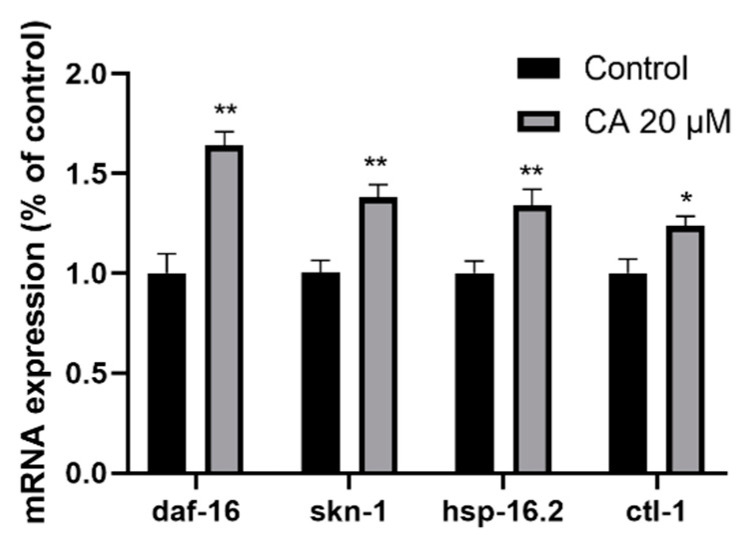
Effect of CA on aging-related gene expression in *C. elegans.* Wild-type worms were treated with DMSO (control) or with 20 μM CA for 3 days, followed by extraction of total RNA and RT-qPCR, as described in the Methods. The relative mRNA expression of *skn-1*, *ctl-1*, *daf-16*, and *hsp-16.2* are presented (*n* = 250–300 worms per group). The difference was examined with student’s *t*-test, * *p* < 0.05, ** *p* ≤ 0.01 vs. control.

**Figure 9 molecules-26-06609-f009:**
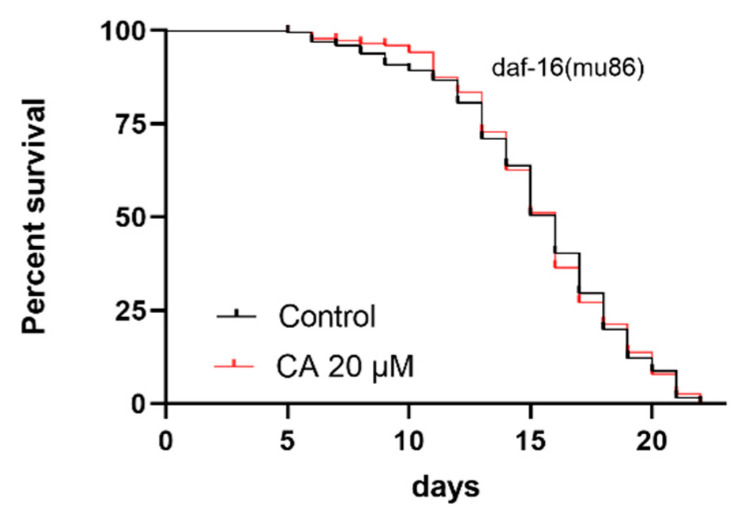
Effect of CA on the lifespan *C. elegans* with a functional mutation in *daf-16.* Worms were raised at 20 °C on NGM OP50 plates without (control) or with CA (20 μM). The CA 20 μM group had almost the same lifespan as the control group (*n* > 216 worms per group, *p* > 0.05).

**Table 1 molecules-26-06609-t001:** Effect of CA on the mean lifespan of *C. elegans*.

	Number of Worms	Mean Lifespan ± SE (Days)	% of Control	*p* Value
Control	161	24.36 ± 0.71	100.00	/
CA 10 μM	146	26.58 ± 0.71	109.11	0.032
CA 20 μM	174	27.05 ± 0.64	111.05	0.023
CA 40 μM	151	25.34 ± 0.72	104.02	0.380

**Table 2 molecules-26-06609-t002:** Effect of CA on the mean lifespan of *C. elegans*.

	Number of Worms	Mean Lifespan ± SE (Days)	% of Control	*p* Value
Control	168	18.92 ± 0.42	100.00	/
Rapamycin	185	20.68 ± 0.42	109.30	<0.05
CA 5 μM	178	20.77 ± 0.40	109.78	<0.05
CA 10 μM	182	21.25 ± 0.41	112.32	<0.05
CA 20 μM	165	22.33 ± 0.46	118.02	<0.0001

**Table 3 molecules-26-06609-t003:** Effect of CA on the mean lifespan of *C. elegans* in the presence of oxidative stress.

	Number of Worms	Mean Lifespan ± SE (Days)	% of Control	*p* Value
Control(paraquat)	86	4.91 ± 0.24	100.00	/
Rapamycin(paraquat)	90	6.22 ± 0.28	126.68	<0.001
CA 5 μM(paraquat)	90	5.34 ± 0.24	108.76	ns
CA 10 μM(paraquat)	90	5.82 ± 0.27	118.53	<0.05
CA 20 μM(paraquat)	89	6.34 ± 0.27	129.12	<0.0001

**Table 4 molecules-26-06609-t004:** Effect of CA on the mean lifespan of *C. elegans* with 37 °C heat shock.

	Number of Worms	Mean Lifespan ± SE (Hours)	% of Control	*p* Value
Control(37 °C)	63	4.67 ± 0.19	100.00	/
Rapamycin(37 °C)	62	5.58 ± 0.22	119.49	<0.05
CA 5 μM(37 °C)	62	5.09 ± 0.21	108.99	ns
CA 10 μM(37 °C)	65	5.31 ± 0.20	113.7-	ns
CA 20 μM(37 °C)	66	5.43 ± 0.22	116.27	<0.05

**Table 5 molecules-26-06609-t005:** Effect of CA on the mean lifespan of CF1038 *daf-16(mu86)*.

	Number of Worms	Mean Lifespan ± SE (Days)	% of Control	*p* Value
Control	216	15.35 ± 0.26	100.00	/
CA 20 μM	232	15.49 ± 0.23	100.91	*p* = 0.9597

## Data Availability

Data are contained within the article.

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
