# Peer review of "Curcumin Acetylsalicylate Extends the Lifespan of Caenorhabditis elegans"

_molecules, 2021, doi:10.3390/molecules26216609_

Round 1
Reviewer 1 Report
The authors demonstrated the effects of antioxidants and longevity by CA in this study. The effects of antioxidants and longevity by CA seem to be an interesting discovery as novel compounds, but the further novelty of the molecular mechanism for this effect appears to be needed.
The authors compared treatment with Rapamycin as a positive control. Have the authors considered the possibility of CA acting similar to that of beneficial effects of Rapamycin? For example, the possibility of antioxidant effect and lifespan extension through autophagy activity, because as the authors mentioned in the discussion section, aspirin has an effect on stress response and lifespan extension along with autophagy activity, it seems likely that CA may act as well.
Second, have the authors observed any differences in feeding behavior by CA treatment? Have the authors considered whether CA contributes to lifespan extension through a mechanism similar to dietary restriction such as eat-2 mutant?
The authors showed that ROS level was reduced by CA treatment. Did the authors consider whether this specifically meant a reduction in mitochondrial ROS? I would recommend the possibility to check whether the decrease in ROS is mainly cytoplasmic or mitochondrial ROS. You can try the measurement of ROS using CellROX green or CellROX red for oxidative stress detection.
And the contents of Fig.1 and Fig.2 seem to have overlap. Also, I wonder why the lifespan of the control A and B in Fig. 1 are different even under the same culture condition.
Looking at the images of SOD-3::GFP and GST-4::GFP in Fig. 6 and Fig. 7, GST-4::GFP shows a significant increase by CA, whereas the image of SOD-3::GFP does not seem to change significantly. Therefore, it is recommended to replace the SOD-3::GFP pictures that match the graph result.
Have the authors considered whether CA-induced lifespan extension is merely lifespan extension, or whether CA treatment also has healthy span extension, ie, improves the locomotion behaviors with lifespan extension?
Authors need appropriate citations for the M&M part.
The manuscript needs to be revised by the format of the journal. In particular, font style and size are not consistent. And it is also necessary to indicate in italics in the gene notation of C. elegans.
Author Response
Please see attached pdf file.

Reviewer 2 Report
The manuscript entitled “Curcumin Acetylsalicylate Extends the Lifespan of Caenorhabditis elegans”, described the effects of a novel compound synthetized by the authors, but in the text neither the structure, nor the synthesis procedure is described. There is a lot in vivo work on the model animal C. elegans to investigate the healthy effect of this compound, however there are several questions that should be addressed.
- English should be thoroughly revised by a Scientific English expert.
- The format should be unified along the text
- “Aspirin” is a trademark name and it comprised an active principal acetyl salicylic acid along with several excipients, therefore the IUPAC nomenclature for the active principle should also be provided
- Lines 62-63 this phrase should be rewritten
- A brief protocol for Curcumin acetylsalicylate synthesis and structures should be provided. Some characterization (HPLC-MS or RMN, absorbance and fluorescence spectra) of it may be useful for the readers. Being named as salicylate is supposed to be neutralized by a base such sodium hydroxide, what is the counter ion? What is the pH of the final solution/plate?
- Gene symbols should be italicized
- C. elegans should be italicized
- Lines 92-103 the paragraph should be rewritten as the synchronization protocol do not make any sense, what is a NGM board, and strips? Eggs do not grow; the eggs hatch and the larvae grow.
- Line 123 Microscope characteristics, as well as the manufacturer should be provided
- Line 129 and others, transferred not transmitted
- Line 136 I understand that all the nematodes in treated with different conditions were transferred to plates supplemented with 5 mM of Paraquat (which also IUPAC nomenclature should be provided), please explain it better because it is not clear in the text.
- Only in one section of Materials and Methods and in Results and Discussion DMSO is used as control as the compound is dissolved in it, why is it? Are the rest of the assays made in water or in DMSO? What about the controls? Because is reported by several studies that DMSO is a DAF-16 and its downstream targets modulator and it increase the lifespan of wild-type nematodes. Thus, the authors should be clear about it in materials and methods, in results, in discussion and in the Figure captions.
- Rapamycin was dissolved in water? And curcumin?
- What was the pH of the aspirin solution? Was the pH adjusted before transferring the worms?
- Lipofuscin experiments were performed with 7 days adult worms, SOD and GST with 3 days worms and DAF localization, ROS measurements and RT-PCR with 5 days worms, why the time differences?
- Question for all the GFP assays, GFP excitation maxima is at 488 nm (blue light) and the emission maximum wavelength is at 510 nm, why use UV excitation (320 nm), the sensibility is lower and it probably causes interferences between lipofuscin and GFP, please explain it.
- Please explain the ImageJ analysis procedure.
- What were the excitation and emission parameters used in the DCFH-DA plate assay?
- Line 152 the fluorescence intensity was measured in the assay? or the worms were catalogued based on the daf-16 localization?
- Figure 1, the experiment was performed with Aspirin or with acetyl salicylic acid. Please include it in materials and methods section. Why “aspirin” was used at 100 µM, curcumin at 20 µM and CA in the range of 10-40 µM? why no use it at the same concentration of 20 µM? “Aspirin” control was performed in DMSO or in water? Because the difference between control 1 and control 2 is 40%, so please explain it.
- Figure 2, please justify the difference between the mean lifespan results obtained in Table 1 and Table 3. Why rapamycin was used at 2 nM, a concentration 10 times lower than CA and curcumin, and 50 times lower that “aspirin”?
- Figure 3 there is a typo in rapamycin spelling. Was DMSO used as control in these experiments? Please indicated it.
- In all the Figures with C. elegans images the scale bar should be present in each image and the scale bar and magnification should be provided in the Figure Caption. Also, the Figures have been taken with much more exposure that necessary thus the autofluorescence of the gut is showing in al the images causing interference.
- Figure 5B, the graph shows the results of the plate analysis or the pixel intensity quantification of the obtained images, please indicate it in the caption. Which conditions are represented as “Blank control”?
- Figure 6, SOD accumulates in the head, the tail and the vulva of the animals and is usually measured in the head to avoid interferences. Why the authors used the images of complete animals?
- Line 341 the authors do not show bioavailability data neither for curcumin nor for the CA compound, thus the improved bioavailability is not verified. Could be the compound be metabolized by the worms?
- Line 428 Please include the reference
Author Response
Please see attached pdf file.

Round 2
Reviewer 1 Report
- Citation of references to lifespan measurements in the M&M part is not appropriate.
- The authors conclude that CA increases healthy lifespan (by showing the reduction in lipofuscin level as a blue channel), however, the previous report (Pincus et al., 2016) strongly suggested that it should be looked at the red channel as a marker or rate of aging. Even if CA reduces red fluorescence, it is necessary to have the conclusion about a healthy lifespan can be obtained both from lipofuscin level and improvement of motility with aging by CA treatment.
- Still, the mechanism for the effect on life span extension by CA needs to be further improved. If the lifespan extension effect of CA treatment is required for the activation of insulin signaling pathway genes such as daf-16, it is necessary to conduct an experiment with mutations in related regulatory genes including daf-2 to have a clear conclusion. In addition, it seems necessary to conduct experiments with control strain and appropriate experimental conditions. In the results of daf-16(mu86) in Fig. 9, the control strain is missing, and the daf-16(mu86) mutant, according to previous reports (Senchuk et al., 2018), showed a similar lifespan to that of N2 at 20°C, however, at 25°C, it had a short lifespan in daf-16(mu86) mutant. Therefore, I think that the effect on the daf-16 activity by CA should be concluded from the results performed at 25 °C with N2 control.
- I still see gene names and species names are not italic.
Reviewer 2 Report
Congratulations to the authors, they have done a really good job improving the manuscrip as requested. therefore I suggest the manuscript to be acepted in current form.
